# Interpretable Machine Learning-Based Influence Factor Identification for 3D Printing Process–Structure Linkages

**DOI:** 10.3390/polym16182680

**Published:** 2024-09-23

**Authors:** Fuguo Liu, Ziru Chen, Jun Xu, Yanyan Zheng, Wenyi Su, Maozai Tian, Guodong Li

**Affiliations:** 1School of Statistics and Data Science, Xinjiang University of Finance and Economics, Urumqi 830012, China; lfg53880@cjc.edu.cn; 2School of Mathematics and Computing Science, University of Electronic Technology, Guilin 541002, China; czr20010714@163.com (Z.C.); 18477195396@163.com (W.S.); 3Department of Mathematics and Data Science, Changji University, Changji 831100, China; mztian@ruc.edu.cn; 4Department of Chemical Engineering, Tsinghua University, Beijing 100084, China; jun-xu@mail.tsinghua.edu.cn (J.X.); zhengyy@mail.tsinghua.edu.cn (Y.Z.); 5School of Statistics, Renmin University of China, Beijing 100872, China; 6Center for Applied Mathematics of Guangxi (GUET), Guilin 541002, China

**Keywords:** three-dimensional printing, interpretive machine learning, SVR, integrated learning, SHAP value

## Abstract

Three-dimensional printing technology is a rapid prototyping technology that has been widely used in manufacturing. However, the printing parameters in the 3D printing process have an important impact on the printing effect, so these parameters need to be optimized to obtain the best printing effect. In order to further understand the impact of 3D printing parameters on the printing effect, make theoretical explanations from the dimensions of mathematical models, and clarify the rationality of certain important parameters in previous experience, the purpose of this study is to predict the impact of 3D printing parameters on the printing effect by using machine learning methods. Specifically, we used four machine learning algorithms: SVR (support vector regression): A regression method that uses the principle of structural risk minimization to find a hyperplane in a high-dimensional space that best fits the data, with the goal of minimizing the generalization error bound. Random forest: An ensemble learning method that constructs a multitude of decision trees and outputs the class that is the mode of the classes (classification) or mean prediction (regression) of the individual trees. GBDT (gradient boosting decision tree): An iterative ensemble technique that combines multiple weak prediction models (decision trees) into a strong one by sequentially minimizing the loss function. Each subsequent tree is built to correct the errors of the previous tree. XGB (extreme gradient boosting): An optimized and efficient implementation of gradient boosting that incorporates various techniques to improve the performance of gradient boosting frameworks, such as regularization and sparsity-aware splitting algorithms. The influence of the print parameters on the results under the feature importance and SHAP (Shapley additive explanation) values is compared to determine which parameters have the greatest impact on the print effect. We also used feature importance and SHAP values to compare the importance impact of print parameters on results. In the experiment, we used a dataset with multiple parameters and divided it into a training set and a test set. Through Bayesian optimization and grid search, we determined the best hyperparameters for each algorithm and used the best model to make predictions for the test set. We compare the predictive performance of each model and confirm that the extrusion expansion ratio, elastic modulus, and elongation at break have the greatest influence on the printing effect, which is consistent with the experience. In future, we will continue to delve into methods for optimizing 3D printing parameters and explore how interpretive machine learning can be applied to the 3D printing process to achieve more efficient and reliable printing results.

## 1. Introduction

Three-dimensional printing, also known as additive manufacturing, is a process of generating a three-dimensional object using a digital file. The 3D printer is equipped with different “printing materials” such as metal, ceramic, plastic, and sand, which serve as tangible raw materials. When connected to a computer, the printer can stack these “printing materials” layer by layer under computer control, ultimately transforming the blueprint on the computer into a physical object. There are various technologies involved in 3D printing, distinguished by the different materials and forming methods used [1]. Common materials used in 3D printing include thermoplastic plastics, metal powders, ceramic powders, edible materials, gypsum materials, aluminum materials, titanium alloys, stainless steel, and rubber-like materials. Based on the different printing materials, 3D printing technologies are generally categorized as fused deposition modeling (FDM), selective laser melting (SLM), digital light processing (DLP), etc. [2].

Additive manufacturing has been growing and has become a pillar in many major industries such as the automotive industry, aerospace industry, and sustainable construction. Most industrial sectors choose to utilize artificial intelligence to increase revenue and reduce working hours, and the additive manufacturing industry is no exception. The application of machine learning (ML) in 3D printing has been a focus of researchers worldwide, mainly aiming to improve the overall design and manufacturing processes, especially in the era of Industry 4.0. It is an emerging technology that optimizes systems by intelligently and efficiently utilizing products, materials, and services. Machine learning in 3D printing can reduce manufacturing time, minimize costs, and improve quality. Currently, ML has a wide range of applications in 3D printing, including design optimization, process improvement, on-site monitoring, cloud 3D printing platforms, and security inspection. ML has proven to be a powerful tool for executing data-driven numerical simulation, design feature recommendation, real-time anomaly detection, and network security [3,4].

In the literature, machine learning has been applied to print design, process optimization [5,6,7,8,9], dimensional accuracy analysis [10,11,12,13], manufacturing defect detection [14,15,16], and material performance prediction [17,18,19,20]. Print design is an important research topic that requires a comprehensive understanding of the capabilities and limitations of 3D printing technology, serving as a critical step in the workflow. ML algorithms are mainly used for feature recognition in print design. For example, Yao et al. designed a hybrid algorithm to recommend additive manufacturing features, using hierarchical clustering to identify the similarity between AM design feature groups and target components. They obtained a tree-shaped diagram that can be pruned into subclusters and trained the SVM classifier using existing industrial application instances. The trained classifier is then used to determine the cutoff line of the tree diagram, which defines the final subcluster containing the recommended AM design features [21]. Additionally, AM encourages the development of new designs, and ML algorithms have been proven applicable in this field, particularly in adjusting material properties and generating new designs. Gu et al. applied machine learning to composite material systems and demonstrated its accurate and efficient prediction of mechanical properties, including toughness and strength, surpassing the existing composite materials in the dataset [22]. Furthermore, during the development of new materials or processes, process optimization is commonly carried out. The characteristics of 3D-printed parts with varying process parameters can be obtained through AM algorithms.

Whether it is the initial print design, feature selection prediction, mid-term process parameter optimization, or troubleshooting and process monitoring during technical operations, the application of ML technology in 3D printing is becoming more mature and can effectively guide actual production. Understanding the relationship between different printing parameters is crucial for optimizing the 3D printing process (extrusion, injection molding, and vat polymerization). However, there is a lack of overall explanation behind the results of machine learning, especially in design optimization, where the identification and selection of optimization parameters lack explanation, including the extent to which scaling adjustments affect printing results. In the study by Jin, ZQ, Zhang, ZZ, et al., a convolutional neural network method was used to detect defects in transparent hydrogel-based bioprinting materials based on layered sensor images and machine learning algorithms, inspired by cooperative game theory. Advanced image processing and enhancement techniques were utilized to detect extracted small image blocks, resulting in high accuracy in anomaly detection. With the prediction of various anomalies, the filling pattern category and location information on the image patches can be accurately determined [23]. Other fields and printing stages rely heavily on ML technology, but there is a significant lack of explanation regarding the application and results of ML technology. Therefore, this paper focuses on the 3D printing of polymers, exploring the influence of material formulation and physical parameters on printing results and providing explanations.

Given the limitations of poor interpretability in the application of ML technology across various fields, and drawing on innovation practices in some disciplines, the literature suggests that the theory of interpretable machine learning is both feasible and logical, particularly with regard to SHAP theory. SHAP is an additive interpretation model constructed by Lundberg in 2017, inspired by cooperative game theory. Its core involves calculating the SHAP values of each feature to reflect their contribution to the predictive ability of the entire model [24]. Building on this theoretical foundation, the ML-SHAP model has become increasingly mature in application, such as the use of XGBoost-SHAP by Dong et al. [25] to explain the relationship between driving behavior and vehicle emission levels, and the use of XGBoost-SHAP by Liao et al. [26] to explain the main factors determining athlete value. The XGBoost-SHAP model has already been applied to quantitatively analyze the contribution of influencing factors.

This paper aims to address the lack of correlation screening and optimization capabilities between driving factors and target variables in previous models, resulting in poor interpretability or even a lack thereof for some influential factors. To do so, we establish an interpretable data-driven model, optimize data combinations, and improve model interpretability. Using 3D printing data obtained from the Polymer Processing Laboratory at Tsinghua University’s Department of Chemical Engineering as an example, we use a combination of machine learning (ML) modules and interpretable SHAP modules to calculate the contribution of each driving factor to the 3D printing result. This aims to provide a basic foundation for identifying and explaining the relevant influential factors in 3D printing. The general research idea of this paper is shown in Figure 1.

## 2. Research Data and Methodology

### 2.1. Description of the Dataset Used

This study establishes an interpretable data model, the ML-SHAP model, to explore the correlation between material formulation and physical performance indicators and printing effects, in order to improve model interpretability. The ML module is coupled with the SHAP module to calculate the contribution of each driving factor to the printing effect, using 3D printing data from the Polymer Processing Laboratory, Department of Chemical Engineering at Tsinghua University as an example to provide a foundation for identifying and explaining relevant influential factors in 3D printing. The evaluation indicators for 3D printing performance recorded in 2018 and 2019 are average length deformation rate, average width variation rate, average thickness variation rate, and average warpage. The indicators for 2020 are spline volume and spline warpage, while those for 2021 are bonding strength, spline volume, and spline warpage. Considering the different printing parameters and evaluation indicators at different times, this paper focuses on analyzing the spline warpage indicator for 3D printing.

The data collected from 2018 and 2020 were consolidated to form the final dataset, including five formulation parameters and physical performance indicators—PLA (%), DR 4468 chain extender (CE), experimental situation of twin-screw blending, die swell ratio, elasticity modulus, and impact strength—as the feature set for machine learning training, and one evaluation indicator, spline warpage, as the label value for the learner. For the performance indicators, here is a brief explanation: Firstly, PLA stands for “Polylactic Acid”, a biodegradable material widely used in 3D printing. In this study, PLA refers not only to the material itself, but also to its percentage content in the printed material, which is denoted as “percentage content of polylactic acid material”. Next, DR 4468 chain extender is a chemical additive specially used in polymer materials, mainly used to improve the molecular weight and intermolecular crosslinking degree of polymers. In 3D printing, this chain extender can enhance the mechanical properties of printed materials, such as toughness and ductility.

Twin-screw extruder: HK26 model, screw diameter 26 mm, length-to-diameter ratio of 40, maximum main unit speed of 600 rpm, manufactured by Nanjing Keya Chemical Complete Equipment Co., Ltd., Nanjing, China.

The die swell ratio (DSR) refers to the cross-section expansion ratio of the material after extrusion mold. This phenomenon is usually related to the rheological properties of the material, especially the elastic recovery of the material after shear and pressure release. In 3D printing, the control of extrusion expansion ratio is very important for printing accuracy and interlayer bonding. By optimizing the printing parameters and material formulation, the extrusion expansion ratio can be adjusted to obtain a more accurate and uniform print layer.

Dual-nozzle industrial-grade 3D printer: UP350 D model, single nozzle build area of 350 mm × 350 mm × 350 mm, dual nozzle build area of 335 mm × 335 mm × 350 mm, manufactured by GuoHang Technology Co., Ltd., China. For this 3D print, we used the self-made filament with a nozzle diameter of 0.4mm, a layer height of 0.2 mm, and a print speed of 50 mm/s. The print temperature was set to about 200 °C, with a bed temperature of 60 °C. The infill was 100% with a grid pattern, and supports were generated automatically. Retraction was set at 6mm with a speed of 40mm/s. The print orientation was optimized for minimal support material, and a heated bed with an adhesive sheet was used for first-layer adhesion. A total of 23 valid data rows were collected from the laboratory data, with a test dataset selected at a ratio of 25%, resulting in 6 and 17 samples for the training and test datasets, respectively. Table 1 shows the input data description collected during the training and testing phases.

As is well known, the correct distribution of input variables can affect the performance of the model. Input variables are composed of main input parameters that strongly affect the 3D printing effect. The total number of data points, minimum value, mean value, standard deviation, maximum value, and percentiles for these input variables are shown in the table below. The 25th, 50th, and 75th percentiles are used to measure the distribution of data. These values indicate the position of a specific percentage of observed values in the data. The 25th percentile, also known as the first quartile (Q1), indicates that 25% of observed values are less than or equal to this value; the 50th percentile, also known as the median, indicates that 50% of observed values are less than or equal to this value; and the 75th percentile, also known as the third quartile (Q3), indicates that 75% of observed values are less than or equal to this value. The total number of data points, minimum, mean, standard deviation, 25%, 50%, 75%, and maximum values for these input variables are described in the table.

In addition, before machine learning training, this paper initially determines the validity of the research by drawing the multiple correlation matrix and provides guidance for the subsequent factor identification. The Pearson correlation coefficient is used to measure the strength of the linear relationship between the two variables. The formula for calculating the correlation coefficient is $r=\frac{\sum \left(x_{i}-\bar{x})(y_{i}-\bar{y}\right)}{\sqrt{\sum {\left(x_{i}-\bar{x}\right)}^{2}\sum {\left(y_{i}-\bar{y}\right)}^{2}}}$; $x_{i}$ and $y_{i}$ are the observed values of the two variables, respectively, and $\bar{x}$ and $\bar{y}$ are their average values. Moreover, this study follows the common practice in statistics and defines the correlation coefficient greater than 0.7 as high correlation. This threshold is based on the consensus of a large number of statistical analyses and studies, among which MAO Shisong also mentioned the calculation and interpretation of correlation coefficients in Probability Theory and Mathematical Statistics [27], which provides us with a theoretical foundation. During the experiment, we collected data on various input variables during the 3D printing process, including PLA content, elastic modulus, and other chemical and physical properties. It was imported into a Python environment, cleaned and standardized, and the above statistical observations were made. The corrcoef function was used to calculate the Pearson correlation coefficient, and the heatmap function of the seaborn library was used to generate the heatmap according to the calculated correlation coefficient matrix. The multiple correlation matrix (heat map) of the studied input and output points is shown in Figure 2. Different colors represent different correlation values.

It can be observed that (1) the correlation between PLA and elasticity modulus with warpage is relatively high, with a correlation coefficient of 80%; (2) there is a high degree of multicollinearity among PLA, elasticity modulus, and warpage; (3) the correlation coefficients between ADR 4468 chain extender, twin-screw blending experiment, and die swell ratio with warpage are not only small but also weakly related to the other three input variables. These results are consistent with previous research findings, where PLA content primarily characterizes the chemical properties of printing materials, while elasticity modulus and impact strength describe physical property features. The chemical properties of printing materials play a decisive role in determining the physical properties of printing results, and physical property features will determine the spline warpage results of 3D printing. Therefore, it is appropriate to use all input variables to improve model accuracy and confirm the impact of each variable on the estimated value of compression strength.

The influence of input variables on 3D printing results, specifically spline warpage, was visualized using a hexagonal contour plot, as shown in Figure 3. The color regions with higher intensity indicate the most useful data points for achieving higher strength characteristics, representing a concentrated range of parameter values that lead to the desired printing outcome in practical experiments. In this study, three parameters with relatively high correlation coefficients were selected from the heatmap: PLA content (0.82), elasticity modulus (0.82), and breaking strength (−0.82). When plotting their relationships with the output parameter using the Seaborn Python package, a contour plot was generated to highlight the desire for smaller spline warpage values. Each input variable has an optimal concentration for achieving this goal. Darker colors indicate that the corresponding spline warpage values are closer to zero under the current parameter settings. It can be observed that each input variable has its own characteristic range of values. Compared to the other two parameters, the range of PLA content is more refined, primarily focusing on the range of 0.6–0.9 for achieving a spline warpage value of zero most effectively. The elasticity modulus ranges from 1000 to 3000 MPa, but values between 1800 and 2200 MPa are more concentrated, resulting in a spline warpage closer to zero and printing outcomes that align better with expectations. Breaking strength is most effective within the range of 3.5–5 kJ/m^2^, leading to experimental conditions that approach perfection with a retention rate of 0–1. The correlation coefficient values between input and output parameters were generated using the Seaborn heatmap function, which creates a heatmap by describing the correlation matrix between inputs and outputs.

### 2.2. Support Vector Regression

When the SVR method identifies the influencing factors of inflation, the function form is $\hat{\pi_{t}}=w^{t}\phi (x_{t})=b$. Where ϕ(⋅) is the basis function for nonlinear transformation of inflation influencing factors, w is the weight of the basis function, and the basis function transformation enables the SVR model to identify the nonlinear relationship between inflation and influencing factors [28,29]. The idea of SVR estimation is to optimize the distance from each sample point to the support vector. This makes the SVR model punitive $L_{2}$ and reduces the negative effects of overfitting and multicollinearity of variables. SVR adds tolerance to error in the loss function, and only when the 3D printing result predicted $\hat{\pi_{t}}$ by SVR deviates from the actual print result $\hat{\pi_{t}}$ by more than a certain degree, the error is recorded in the loss function. Objective optimization function of SVR:(1)$$\begin{array}{l} \underset{w,b}{\arg\min}\frac{1}{2}{\left\|w\right\|}^{2}+C\sum_{i=1}^{N}\left(\xi_{i}+\xi_{i}^{*}\right)\\ s.t\left\{\begin{array}{l} \pi_{t}-w^{T}\phi (X_{t})-b\leq \varepsilon +\xi_{t}\\ \pi_{t}-w^{T}\phi (X_{t})-b\geq -\varepsilon -\xi_{t}^{*}\\ \xi_{t}\geq 0,\xi_{t}^{*}\geq 0 \end{array}\right. \end{array}$$

### 2.3. Integration Method Based on Regression Tree

When the method based on regression tree is used to identify the influencing factors of 3D printing parameters, the function form is $\hat{\pi_{t}}=\sum_{i}\bar{\pi_{i,t}}\cdot I(X_{t}\in R_{i})$. According to the relevant influencing factors, the sample space i is divided into different regions $R_{i}$, and the mean value of the predicted region is taken as the predicted value of the sample falling into the region. The region divided by regression tree implies the relationship between various influencing factors and the volume retention rate, especially the influence of the interaction between influencing factors on the printing effect [30]. The three methods selected in this paper, RF, gradient boost, and XGBoost, are all ensemble learning methods based on regression trees, derived from the ensemble ideas of bagging and boosting [31,32]. Although the calculation process of the three methods is different, the recognition of the relationship between print parameters and volume retention rate is essentially similar to the regression tree method.

The RF method is based on the bagging integration method. Bagging integration method integrates the prediction results of multiple regression trees, which effectively improves the prediction effect of the model. Specifically, bagging [33,34] trains multiple machine learning models by constructing multiple different training samples through resampling, and takes the mean predicted by all models as the final predicted value. The RF model is improved on the basis of bagging. When the tree model in RF splits nodes, it randomly selects some factors affecting inflation as nodes to divide regions, thereby reducing the similarity between each model and improving the overall forecasting effect and robustness of the model [35].

Gradient boost and XGBoost are based on the boosting integration method. Boosting’s idea is to improve the model’s predictive performance by training multiple models several times and integrating their results. Gradient boost and XGBoost [36] are both improvement methods based on the boosting idea. In each iteration estimate, the gradient boost new model will fit the prediction residual of the existing integrated model, and finally minimize the overall prediction loss function [37,38].

In this paper, supposing that the regression tree model of the m-th iteration of gradient boost is $h_{m}(X_{t})$, the loss function is $L(\pi_{t},{\hat{\pi }}_{t,m-1}(X_{t})+v\cdot \beta_{m}h_{m}(X_{t}))={\left(\pi_{t}-{\hat{\pi }}_{t,m-1}-v\cdot \beta_{m}h_{m}(X_{t})\right)}^{2}$, ${\hat{\pi }}_{t,m-1}(X_{t})$ is the volume retention rate predicted by the integrated model based on the sample $X_{t}$ of influencing factors after the *m* − 1-th iteration, and v is the learning rate of the regularization model. The hyperparameters of this model include the number of regression trees, the depth of each regression leaf node, the learning rate, etc., and their optimal values can be obtained through certain methods; the specific methods are described below. XGBoost improves the training efficiency of gradient boost, utilizes the second-order information of the loss function, and adds the penalty term of model complexity to the loss function L. In this paper, the loss function optimized by XGBoost is as follows:(2)$$L_{m}=\sum_{t=1}^{T}g_{tm}h_{m}(X_{t})+\frac{1}{2}S_{tm}h_{m}^{2}(X_{t})+\Omega (h_{m})$$
where $g_{tm}$ is the first derivative and $S_{tm}$ is the second derivative with respect to the loss function ${\hat{\pi }}_{t,m-1}(X_{t})$, respectively, and the penalty term is as follows:(3)$$\Omega (h_{m})=\alpha W_{m}+\frac{\lambda }{2}{\left\|w_{m}\right\|}^{2}$$
to punish the norm of the middle node of the regression tree.

### 2.4. Feature Importance

The purpose of feature importance is to rapidly identify and compare the most significant factors, but its limitations include the difficulty in determining the positive or negative influence of a factor on the model results and the lack of consideration for interaction effects. Generally, feature importance can be obtained using the feature_importances_ method [39] of the model.

The feature_importances_ in xgboost, for instance, can be calculated either by the number of times a feature is split or by the gain obtained from splitting on that feature. This is also one of the advantages of using gradient boosting, as it allows for convenient retrieval of importance scores for each attribute after constructing the boosted trees. The underlying idea is that the significance of a feature lies in its ability to reduce the uncertainty of the prediction target. Features that can more effectively reduce this uncertainty are considered more important. In other words, the calculation of feature importance is based on the mean decrease impurity, which measures the information gain (Gini coefficient) before and after splitting on a particular feature during the decision tree construction process.

In the context of this study, machine learning methods are applied to predict the effects of 3D printing. Similarly, the contribution of features during the construction of decision trees is observed. Typically, feature importance provides a score indicating the usefulness or value of each feature in building the boosted decision trees of the model. The higher the number of crucial decisions made by an attribute for the decision tree, the higher its relative importance. This importance is explicitly calculated for each attribute in the dataset, allowing for ranking and comparison of attributes. The importance of an individual decision tree is computed by the number of times the attribute improves the performance metric at each split point, weighted by the number of observations handled by the node. The performance metric can be a measure of purity (such as the Gini coefficient [40]) used for selecting split points or other more specific error functions. Finally, the feature importances are averaged across all decision trees in the model to obtain concrete values for identifying the factors influencing the outcome.

### 2.5. Shapley Additive Explanation

The SHAP value interpretation method was proposed by Lundberg and Lee [24] to explain the contribution of each influencing factor in the machine learning model to the target predicted value. SHAP belongs to the method of model post-interpretation, its core idea is to calculate the marginal contribution of features to the model output, and then explain the “black box model” from the global and local levels. SHAP builds an additive interpretive model where all features are considered “contributors”. For each predicted sample, the model produces a predicted value, and SHAP value is the value assigned to each feature in that sample.

The set F represents the total set or feature set of all influencing factors, and the elements in it are called features and are denoted $\left|F\right|$ as the number of elements in feature set F. The predicted value $\hat{f}(x_{*})$ of a machine learning model f(·) at a particular sample point $x_{*}$ can be decomposed into the following:(4)$$\hat{f}(x_{*})=\emptyset_{0}+\sum_{i=1}^{\left|F\right|}\emptyset_{*}^{j}$$
where $\emptyset_{0}$ is the base value of the model prediction, generally the forecast mean value $E[\hat{f}\left(x\right)]$ as the base value. $\emptyset_{*}^{j}$ is the size of the influence of the *j*-th influencing factor on the prediction on the sample $x_{*}$, i.e., the SHAP value. The larger the value $\left|\emptyset_{*}^{j}\right|$ is, the greater the influence of the influencing factor on the predicted value of the target. And $\emptyset_{*}^{j}>0$ and $\emptyset_{*}^{j}<0$, respectively, indicate that the influencing factors have a positive or negative impact on the predicted value. The SHAP value $\emptyset_{*}^{j}$ of the *j*-th factor is calculated by the following formula:(5)$$\emptyset_{*}^{j}=\sum_{s\subseteq F\backslash \{j\}}\frac{\left|S\right|!(\left|F\right|-\left|S\right|-1)!}{\left|F\right|!}(v_{*}(S\cup \{j\}-v_{*}(S))$$
where S is the set that does not contain the *j*-th factor (a subset of the set F), given the number of elements of the subset $\left|S\right|$. Based on the feature set F, it can build a total of subsets that do not contain the j-th factor, whose number is $\sum_{S\subseteq F\backslash \left\{j\right\}}\frac{\left|F\right|!}{\left|S\right|!\left(\left|F\right|-\left|S\right|-1\right)!}$. $\left(\nu_{*}(S\cup \left\{j\right\}-\nu_{*}(S)\right)$ is the expected influence of factors j on the predicted value under the condition that the combination of control variables is a subset.

The calculation of the SHAP value in this paper is based on the trained machine learning model, so the calculation of the SHAP value of each factor according to Aas et al. [41] still uses the information of all training samples when calculating SHAP value of each sample. The specific research ideas and calculation steps are as follows.

Assume that the sample set is $x=\left\{x_{1},\ldots ,x_{n}\right\}$, and a 3D printing experiment has been carried out. The SHAP value $\emptyset_{i}^{j}$ of the *j*-th factor on the sample is calculated according to the method, that is, the influence of the factor j on the 3D printing effect. Based on the SHAP value of the factor j in the sample $x_{i}$, the SHAP value sequence is $\emptyset^{j}=\left\{\emptyset_{1}^{j},\emptyset_{2}^{j},\ldots ,\emptyset_{n}^{j}\right\}$, which reflects the influence of the factor on the whole sample set, and the contribution size comparison of each feature is obtained in the set.

### 2.6. Models Performance Measurements

In model performance evaluation, it is important to implement uncertainty analysis to evaluate the performance of machine learning models [42]. Using the training dataset to construct the prediction model, test datasets are used to evaluate the accuracy of the model. Six statistical evolution measures, including coefficient of certainty (R2), explained_variance_score (explained_variance_score), root mean square error (RMSE), and mean absolute error (MAE), are proposed to evaluate the performance of the model [43,44]. These performance metrics are good indicators for evaluating overall prediction accuracy, with their mathematical expressions and typical ranges of accuracy shown in Table 2.

## 3. Results and Discussion

### 3.1. Model Fitness Analysis Based on Machine Learning Prediction

#### 3.1.1. Determination of Model Parameters

After data preprocessing and feature engineering of the original dataset, the important hyperparameters of each model need to be determined. According to previous experience, the hyperparameters of support vector regression are determined by Bayesian optimization [45], while the other three regression tree-based integration methods use grid search to determine the hyperparameters.

Among them, the hyperparameters of support vector regression (SVR) mainly include penalty coefficient C and kernel function, which is the inner product of two samples transformed by the basis function. There are many kinds of kernel functions, such as linear kernel, polynomial kernel, sigmoid kernel, and RBF (radial basis function) kernel [46]. After searching by the Bayesian optimization algorithm, the penalty coefficient C is determined to be 5, and the kernel function is selected as RBF. The RBF kernel can map the sample to a higher-dimensional space and can handle the sample when the relationship between class labels and features is nonlinear. The formula is as follows:(6)$$K(x,y)=e^{-\gamma ||x-y|{|}^{2}}$$

The regression tree-based integration methods (random forest, gradient lift, and extreme gradient lift) are similar, and the common important parameters [47] are max_depth (maximum depth of the tree), learning_rate (learning rate), and n_estimators (number of base evaluators). In addition, there is another important hyperparameter in GDBT and XGBoost—subsample (the proportion of samples sampled)—but the quantity of data in this study is small, so the default value 1 is selected. On this basis, compared with gradient boost, the hyperparameter of the XGBoost model has two more penalty terms: reg_alpha (the weight of L1 regular term) and reg_lambda (the weight of L2 regular term). In this paper, some hyperparameters are selected, and the optimal combination of hyperparameters of the three methods is shown in the following Table 3:

#### 3.1.2. The Fitting Results of Four Machine Learning Methods

Before the empirical analysis, it is necessary to compare the prediction performance of the four machine learning methods selected in this paper. It is common practice to select explained_variance_score of the training set and mean square error (MSE) and mean absolute percentage error (MAE) of the test set as the evaluation indexes of the model prediction performance.

This article uses Python 3.10, using the sklearn package comprising SVR, RandomForestRegressor, the GrandientBoostingRegressor function, and the Xgboost XGBRegressor function. Among them, the hyperparameter setting of specific functions will be explained in detail below. In addition, train_test_split was used to conduct test set and training plans for the data. The sample size was 25, and the ratio of the two categories was 1:3, with 718 samples.

For the fitting results of machine learning, this paper first draws a discounted comparison graph between the predicted value and the real value of the training set and adds an error bar graph for more intuitive observation. Shown in Figure 4a–d, respectively, are the proxy models fitted by SVR, RF, GBDT, and XGB models to predict “volume retention rate”, in which the horizontal axis represents the sample points, the left vertical axis scales the errors and is drawn as a column chart, and the right vertical axis is the volume retention value and is drawn as two line charts.

The outcomes show that there is a clear difference between the predicted and true values of the SVR model. The predicted value always remains at a fixed value, while the real value shows great volatility. This shows that the SVR model has a large error in the prediction of “volume retention rate” and cannot accurately capture the difference between samples. The possible reason is that the SVR model does not consider the nonlinear relationship between samples in the fitting process, which leads to the unsatisfactory fitting effect. In contrast, the RF, GBDT, and XGB models show a better fit in Figure 4b–d. The curve between the predicted value and the true value is similar, and the error is small. This shows that the integrated method model with decision tree as the underlying logic has higher generalization ability and accuracy in predicting “volume retention rate”, and these three models can better capture the change rule of “volume retention rate” and make accurate prediction. This may be because these models are able to deal with nonlinear relationships and have a strong ability to fit. Some degree provides an important reference for us to choose the appropriate machine learning model and provides guidance for further improving the prediction model.

Based on the observation of the fitting chart of the training set, this paper further analyzes the rating index results of the test set. From Table 4 above, we find that by adjusting the parameters, the machine learning model on the whole has a very good performance in predicting the 3D printing effect, with R square reaching more than 80%. This paper describes the importance relationship between variables on the basis of the prediction model, and it is necessary to improve the degree of interpretation and reduction of the model as much as possible. The model shows almost perfect accuracy and describes the relationship between the predicted data well. On the basis of these models, it is reliable to investigate the influence degree of each feature. Secondly, XGBoost and GBDT have the best performance, the R square of the former is nearly 100%, and the MAE and RMSE of the latter are close to 0, which greatly learn the relationship between features and labels, and focus on the recognition of influencing factors after the two proxy models. In addition, it is not difficult to find that the machine model indexes of the last three regression integration trees are better than those in SVR; specifically, the RMSE and MAE of the test set are smaller, and the R-square level is higher. In particular, considering that during the construction of the tree model, features will be evaluated and screened to determine nodes for splitting, which will be of great reference and help in the subsequent evaluation of influencing factors, this paper considers the observation and analysis of the feature scores inherent in the tree model in the next section.

### 3.2. Print Factor Recognition Based on Interpretative Machine Learning Method

Based on the established prediction model mentioned above, we found that SVR performs slightly worse compared to ensemble trees based on decision trees. Therefore, considering the principles of node splitting in the tree construction process from the three decision tree algorithms, we aim to compare the importance of different features.

This article uses Python 3.10, with the aid of the RandomForestRegressor, GrandientBoostingRegressor function, and Xgboost XGBRegressor function of the sklearn package, to output the feature_importance variable. The program draws a series of importance score bar charts—the horizontal axis is the importance score, the vertical axis is the feature, in which the characteristics “PLA”, “PBS”, “ADR 4468 chain extender”, “elastic modulus”, “breaking strength”, “elongation at break”, and “impact strength” are named “f0”, “f1”, “f2”, “f3”, “f4”, “f5”, and “f6”; the results are shown in Figure 5 below.

For the SHAP interpretation method, this paper uses the SHAP package in Python 3.9. SHAP (Shapley additive explanation) is a “model explanation” package developed by Python [48] that can explain the output of any machine learning model with the following variables: shap_values is used to represent the SHAP values of the factors in the model on the samples, so as to draw the bar chart according to it for easy observation and compare with the importance bar chart of the function. The horizontal axis is mean_shap_value, that is, the SHAP value, and the vertical axis is the characteristic, wherein the characteristics of “PLA”, “PBS”, “ADR 4468 chain extender”, “elastic modulus”, “breaking strength”, “elongation at break”, and “impact strength” are named feature 0–feature 6. The results are shown in Figure 6 below.

Looking at Figure 5, we can see a more obvious pattern. The three methods show a fairly consistent arrangement of importance: feature 4—elastic modulus; feature 0—PLA; and feature 5—impact strength are highlighted to have a large impact on the print effect–spline warpage. The evaluated values are also similar. Therefore, in summary, the elastic modulus maintains the highest contribution, followed by PLA and impact strength, which have a greater influence on sample warpage. This conclusion is confirmed by the correlation coefficient of the above thermal map; the chemical properties of the printing material play a decisive role in its physical properties, and the physical characteristics will also determine the spline warping results of 3D printing, although different models have slightly different assessments of the contribution of these two aspects, but they can identify the importance of PLA and elastic modulus. The application of interpretative machine learning to the influencing factors of 3D printing parameters is further explored by observing the evaluation situation under the SHAP value method, as shown in Figure 6 below.

Firstly, the results of each method are compared horizontally. The sorting conclusions obtained by the feature_importance method and the SHAP value calculation theory are highly consistent, which again proves the rationality and credibility of the evaluation results. Secondly, the vertical comparison of the estimation results of the three methods on the importance of influencing factors shows that the histogram is arranged from the highest to the bottom according to the importance scores. There are some differences in the results of the three methods, but they also show more common points. Among them, the three methods consider that the characteristic f5—elongation at break has the highest contribution value, and even though GBDT is evaluated as the second, it closely follows the first contribution variable. In addition, the characteristic f4—breaking strength is also at the top of the score, and the characteristic F0—PLA, f1—PBS, and f2—chain extender ADR4468 have a relatively consistent judgment, which thinks that the contribution is weak and has no great impact on the printing effect. In addition, the evaluation of f3—impact strength and feature 6—impact strength is slightly uneven. Random forest and XGB rank them higher, ranking them in the first three teams. However, the SHAP values of specific observation are close to 0.03–0.05. Similarly, for impact strength, GBDT is different from the other two methods, it is considered to contribute the third, and the SHAP value reaches 0.05, while the other two methods evaluate 0.02, but on the whole, the contribution rate of SHAP value is similar. Taking into account the performance comparison of prediction models, the XGBoost model has relatively high reliability and interpretation. In general, elongation at break, fracture strength, elastic modulus, and impact strength have a deep impact on the 3D printing effect, and the degree is weakened in turn. The importance scores of specific features are shown in Table 5 below.

## 4. Conclusions

This study applied successful machine learning prediction methods to explore the influencing factors of 3D printing in the field of polymers. Despite a limited dataset, the models achieved high accuracy in capturing hidden relationships and establishing a solid foundation for future feature impact assessments. Considering the nonlinear relationships between 3D printing-related features and printing outcomes, this study not only employed classical and efficient machine learning methods to characterize the relationships between features and labels but also utilized interpretable machine learning methods to provide comprehensive explanations for complex relationships that cannot be described by explicit mathematical functions. The methodologies employed in this study can be applied to more advanced black-box models, offering valuable insights for subsequent research. The main research findings are as follows:Through correlation coefficient analysis, we found that among the input variables, the PLA content and elastic modulus showed the highest correlation with warpage, with a correlation coefficient of 80%. There was also a high degree of multicollinearity between PLA content, elastic modulus, and warpage. On the other hand, there was a weak correlation between ADR 4468 crosslinking agent, twin-screw blending, extrusion swell ratio, and both warpage and the other three input variables.In terms of model selection, we employed three machine learning algorithms, namely, gradient boosting decision trees (GBDTs), random forest (RF), and support vector regression (SVR), to predict “spline warpage,” achieving satisfactory results. It is worth noting that these results were obtained through debugging using a small dataset, yet these models demonstrated good generalization capabilities and can be applied to larger-scale datasets.Additionally, we introduced the SHAP (Shapley additive explanations) interpretable machine learning framework to explain the predictions of the models. Through SHAP value analysis, we discovered that fracture elongation, fracture strength, elastic modulus, and impact strength have significant impacts on 3D printing outcomes, with the influence decreasing in that order. This conclusion is consistent with practical experience and aligns with our preliminary finding that chemical properties affect physical features, which, in turn, determine printing outcomes.

However, it should be noted that in constructing the prediction model for 3D printing outcomes, this study treated indicators such as bonding strength and spline volume as individual labels for machine learning models, training and examining them separately. The study did not consider the interaction effects between parameters, which may introduce subjectivity. Furthermore, given the continuous advancements in experimental equipment and materials, the input features of the 3D printing experimental prediction models may vary over time. This study focused on individual years (2018, 2019, 2020, and 2021) without aggregating and analyzing the results, potentially missing out on underlying physical laws. These limitations require further research to overcome.

## Figures and Tables

**Figure 1 polymers-16-02680-f001:**
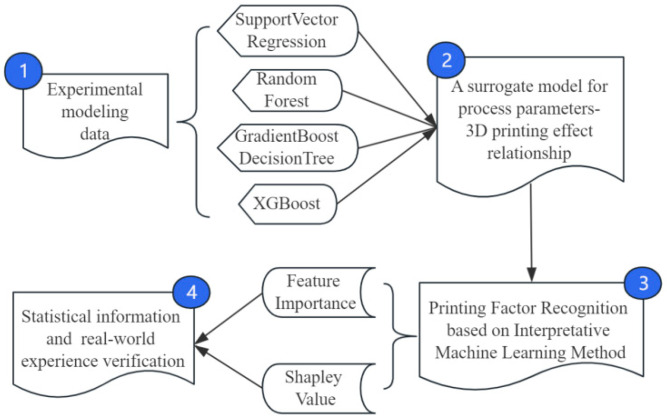
The outline of the proposed framework.

**Figure 2 polymers-16-02680-f002:**
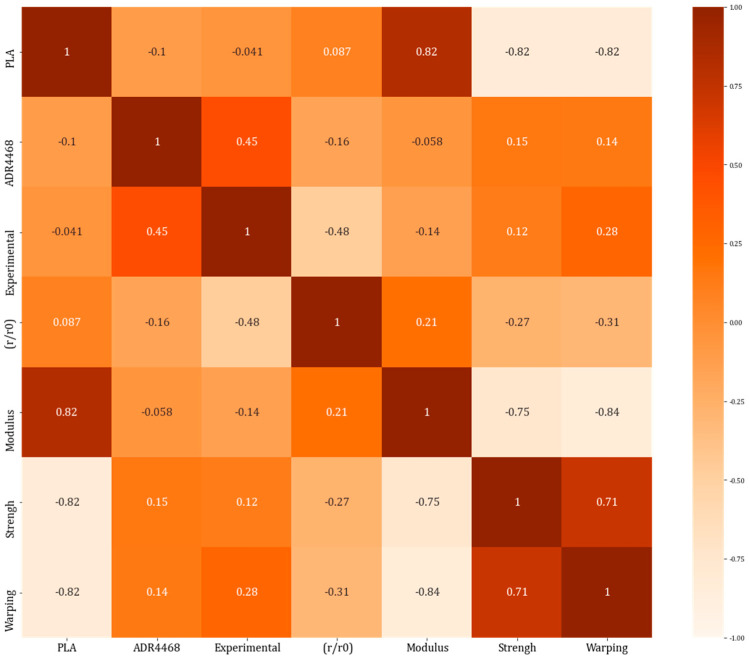
Thermal map of correlation coefficients between print parameters and spline warpage.

**Figure 3 polymers-16-02680-f003:**
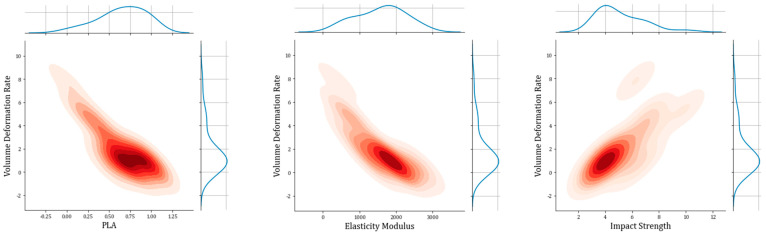
The hexagon contour plot of parameters and spline warpage.

**Figure 4 polymers-16-02680-f004:**
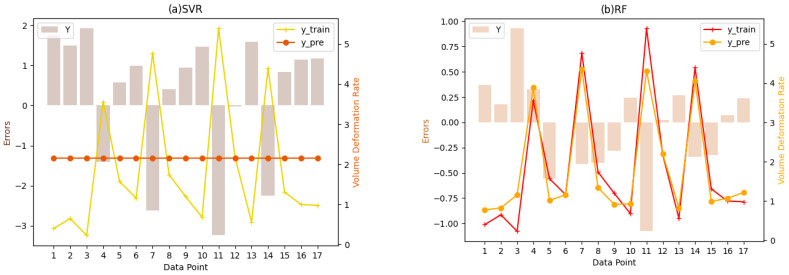
Prediction performances of the rate of volume retention by machine learning models. Subgraphs (**a**–**d**) are the proxy models for predicting “volume retention rate” fitted by SVR, RF, GBDT, and XGB models, respectively, where the horizontal axis is the sample points, the left vertical axis is the errors, the right vertical axis is the volume retention value, and the volume retention rate is drawn as two line graphs.

**Figure 5 polymers-16-02680-f005:**
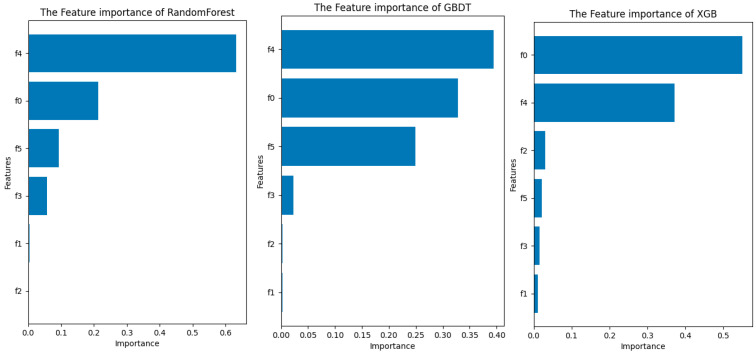
Three-dimensional printing impact factors assessment columnar comparison chart.

**Figure 6 polymers-16-02680-f006:**
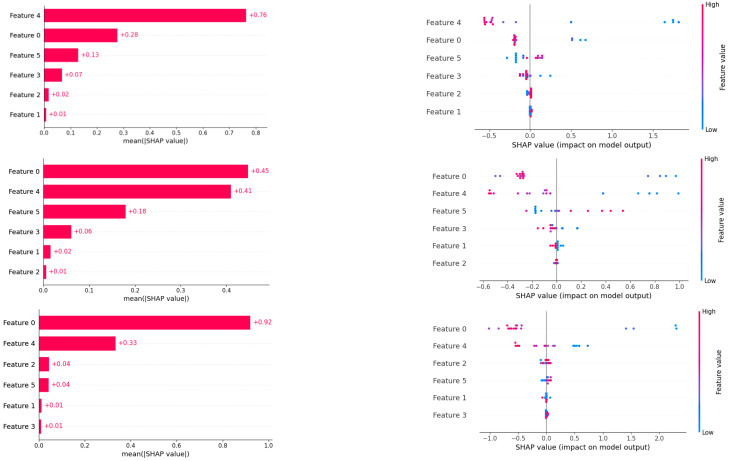
A columnar comparison chart of the factors influencing the evaluation of the integrated approach model.

**Table 1 polymers-16-02680-t001:** The statistics table of spline warpage.

	PLA (100%)	Chain Extender ADR4468 (100%)	Twin-Screw Extrusion Experimental Conditions	Die Swell Ratio (r/r0) (100%)	Elastic Modulus (GPa)	Impact Strength (kJ/m^2^)	Warping (100%)
Count	23	23	23	23	23	23	23
Mean	0.652	0.002608696	3.695652174	1.689130435	1584.827391	5.11821739	1.8786957
Std	0.279	0.00255377	1.329209697	0.512559647	654.5020297	1.68434678	1.9694007
Min	0	0	1	1	392.8	3.334	0.23
0.25	0.5	0	2.5	1.275	1110.065	3.858	0.61
0.5	0.7	0.005	4	1.8	1657	4.477	1.16
0.75	0.85	0.005	5	2	2015.395	6.1065	1.96
Max	1	0.005	5	3.3	2708.84	9.94	7.8

**Table 2 polymers-16-02680-t002:** The evaluation index table of fit degree of machine learning model.

Assessment Criteria	Standard Range
$R^{2}(y,\hat{y})=\frac{\sum_{i=1}^{n}{\left(y_{i}-{\hat{y}}_{i}\right)}^{2}}{\sum_{i=1}^{n}{\left(y_{i}-\bar{y}\right)}^{2}}$, $y_{i}$: observed data, ${\hat{y}}_{i}$: predicted data, and $\bar{y}$ is the mean	0 to 1
$Explained\_\mathrm{var}iance\_score(y,\hat{y})=1-\frac{\mathrm{var}(y_{i}-{\hat{y}}_{i})}{\mathrm{var}(y_{i})}$, $y_{i}$: observed data, ${\hat{y}}_{i}$: predicted data	0 to 1
$MAE(y,\hat{y})=\frac{1}{n}\sum_{i=1}^{n}\left|y_{i}-{\hat{y}}_{i}\right|$, $y_{i}$: observed data, ${\hat{y}}_{i}$: predicted data and n is the number of observations	0 is the best value
$RMAE(y,\hat{y})=\sqrt{\frac{1}{n}\sum_{i=1}^{n}\left|y_{i}-{\hat{y}}_{i}\right|}$, $y_{i}$: observed data, ${\hat{y}}_{i}$: predicted data, and n is the number of observations	0 is the best value

Using training dataset to construct prediction model, test datasets are used to evaluate the accuracy of the model. Six statistical evolution measures, including coefficient of certainty (R^2^), explained_variance_score (explained_variance_score), root mean square error (RMSE), and mean absolute error (MAE), are recommended to evaluate the performance of the mode.

**Table 3 polymers-16-02680-t003:** The main parameter configuration table of the forecast “volume retention rate” model.

Model Name	Parameter Configuration
SVR	C = 4.9284, Kernel = RBF
Random Forest	max_depth = 3, max_features = 5, n_estimators = 422
GBDT	max_depth = 3, max_features = 4, n_estimators = 18
XGBoost	max_depth = 2, n_estimators = 18, reg_lambda = 1.4423

**Table 4 polymers-16-02680-t004:** Evaluation index results of four machine learning algorithms.

	SVR	RF	GBDT	XGB
$R^{2}$	0.8096	0.8498	0.9369	0.9794
explained_variance_score	0.8179	0.8509	0.9377	0.9794
MAE	0.3367	0.2364	0.0556	0.2742
RMSE	0.2026	0.1038	0.0043	0.1919

**Table 5 polymers-16-02680-t005:** Feature importance characteristic score.

	RF	GBDT	XGB
PLA	0.212644311	0.328358355	0.5503053
ADR 4468 chain extender	0.00365122	0.002253861	0.010976699
Modulus of elasticity	0.002183382	0.002425572	0.030449962
Breaking strength	0.056911609	0.023243075	0.015605606
Elongation at break	0.631832284	0.393985351	0.37159628
Impact strength	0.092777195	0.249733786	0.021066085

## Data Availability

Data are contained within the article.

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
