# Peer review of "Interpretable Machine Learning-Based Influence Factor Identification for 3D Printing Process–Structure Linkages"

_polymers, 2024, doi:10.3390/polym16182680_

Round 1

Reviewer 1 Report

Comments and Suggestions for Authors

Review report

I have gone through the manuscript titled “Interpretable machine learning-based influence factor identification for 3D printing process-structure linkages” by Liu et al.

The authors analyze the effect of the process parameters on the efficiency of 3D printing process, using SVR, random forest, GBDT and XGB machine learning techniques. The manuscript is well written and the results are presented clearly. I recommend the manuscript for publication after a minor review with respect to the following points.

1) Please expand SVR, GBDT and XGB in the abstract.

2) The last line of the abstract: “In the future ..” is not required and may be moved to the end of conclusion.

3)  Page 2, line nos. 68-70: reference is missing for Yao et al. Please check. Bring the reference from Line 74 to earlier.

4) Line 79: “machine learning ..” is not required. Move reference 22 to the end of the earlier sentence.

5) The introduction section is too long. The 2nd and 3rd paragraphs can be shortened and combined, retaining the essential information only.

6) Figure 1 is not mentioned in the text. Please check.

7) Line 267-270: please remove the underscores.

8) Line 378: change “USES” to “uses”.

9) Please use arrow to show the correspondent axes in Fig. 4(a-d).

Comments on the Quality of English Language

Minor editing is required. Can be done during proof stage.

Author Response

Comments 1: Please expand SVR, GBDT and XGB in the abstract.

Response 1: Thank you for pointing this out. I agree with this comment. Therefore, I have adjusted this, marked in red on line22-34.

Comments 2: The last line of the abstract: “In the future ..” is not required and may be moved to the end of conclusion.

Response 2: Thank you for pointing this out. I agree with this comment. So, I deleted 'in the future.'

Comments 3: Page 2, line nos. 68-70: reference is missing for Yao et al. Please check. Bring the reference from Line 74 to earlier.

Response 3: Thank you for pointing this out. I should not have omitted the literature on Yao.et al. The main reference information about it is in line 87.

Comments 4: Line 79: “machine learning ..” is not required. Move reference 22 to the end of the earlier sentence.

Response 4: Thank you for pointing this out. I agree with your suggestion. That sentence is not smooth enough and has been modified in red at line 93 of the article.

Comments 5: The introduction section is too long. The 2nd and 3rd paragraphs can be shortened and combined, retaining the essential information only.

Response 5: Thank you for pointing this out. We believe that this article is about the cross-study of artificial intelligence and polymer materials, and it is necessary to make a full explanation of the relevant background and research situation. Thank you for your suggestions.

Comments 6: Figure 1 is not mentioned in the text. Please check.

Response 6: Thank you for pointing this out. I agree with this comment.  Therefore, I add a sentence in line 137-138 of the article, marked in red.

Comments 7: Line 267-270: please remove the underscores.

Response 7: Thank you for pointing this out. I have removed these.

Comments 8: Line 378: change “USES” to “uses”

Response 8: Thank you for pointing this out. I have changed it on Line 398, 458.

Comments 9: Please use arrow to show the correspondent axes in Fig. 4(a-d).

Response 9: Thank you for your review of our manuscript and your valuable suggestions. We have given careful consideration to your suggestion to add an arrow identifier for the axis in Figure 4 (a-d). We believe that the current chart design is clear enough, and the identification and legend of the axes are sufficient to help the reader understand the data. Our goal was to keep the charts concise and avoid excessive visual elements that might distract the reader.

However, we also value your professional advice. If the readers of your issue generally agree that adding arrows would improve the readability of the chart, we are willing to reconsider. Do you have any other suggestions or further guidance that could help us improve the presentation of the chart?

Reviewer 2 Report

Comments and Suggestions for Authors

This study focuses on predicting the impact of 3D printing parameters on the printing effect by using machine learning methods. The 20 influence of the print parameters on the results under the feature importance and shap values is 21 compared to determine which parameters have the greatest impact on the print effect. Overall, this paper presents high novelty which can be published in Polymers journal. Some of the minor comments are given below, addressing which, would attract more readers and researchers:

1. In Line 101, the authors have used the term ‘material formulation’. Does that mean, the material composition?

2. Briefly describe the spline warpage which is the main indicator of this study.

3. Line 143, what is DR 4468 chain extender (CE)? Kindly elaborate on it. Also, elaborate twin-screw blending and die swell ratio. Please mention how this is relevant to 3D printing technology.

4. In section 2.1, the authors should briefly describe the volume retention rate

5. In Line 151, the authors have repeatedly mentioned about the input variables that can affect the performance of the model. However, they have not mentioned which are those input variables.

6. Figure 1 shown in the manuscript is very generalized. If possible, try to make it more specific as per this study so that readers can quickly get the insight. For example, mention the parameters included and briefly describe the significance of each step.

7. In Table 1, what does PLA signify? Does it mean % of PLA material??

8. Kindly include the units for all the parameters in Table 1.

9. The methodology employed in generating the thermal map of the correlation coefficient has to be explained in detail. How the correlation between the PLA and warpage are determined? Is it by analyzing the experimental data?

10. In Line 186, the hexagonal contour plot, as shown in Figure 2 should be corrected to Figure 3.

11. The authors should use specific terminologies for readers to better understand without confusion. I can see that the impact strength is also termed as breaking strength, Spline warpage as volume deformation rate. Please use a single terminology for each parameter. 

Comments on the Quality of English Language

Minor English editing is required. It is suggested to use Grammerly software to correct the minor mistakes.

Author Response

Comments 1:   In Line 101, the authors have used the term ‘material formulation’. Does that mean, the material composition?

Response 1:  Yes, you understand correctly. The "material formulation" mentioned in the paper does refer to the composition, combination and approximate dosage of materials. The term is commonly used to describe a detailed list of all the different ingredients that make up a material, including the type and proportion of each ingredient, information that is essential for copying or recreating the material. Yes, you understand correctly. The "material formulation" mentioned in the paper does refer to the composition, combination and approximate dosage of materials. The term is commonly used to describe a detailed list of all the different ingredients that make up a material, including the type and proportion of each ingredient, information that is essential for copying or recreating the material. Thank you for your attention to this term, and we have ensured that "material formulation" is clearly defined and explained in the paper to avoid any possible misunderstandings. Thank you for your attention to this term, and we have ensured that "material formulation" is clearly defined and explained in the paper to avoid any possible misunderstandings.

Comments 2:  Briefly describe the spline warpage which is the main indicator of this study.

Response 2:  Thank you for your question. I agree with this comment. Relevant definitions have been added on lines 183-189 of the article and are marked in red.

Comments 3:   Line 143, what is DR 4468 chain extender (CE)? Kindly elaborate on it. Also, elaborate twin-screw blending and die swell ratio. Please mention how this is relevant to 3D printing technology.

Response 3:  Thank you for your question. I agree with this comment. Relevant definitions have been added on lines 169-182 of the article and are marked in red.

Comments 4:  In section 2.1, the authors should briefly describe the volume retention rate.

Response 4:  A common improvement has been made with the review suggestion of Article 11, and the problems are that there are various expressions of concepts in this paper. the volume retention rate has been uniformly changed to Spline warpage, because this is the only label of the learner studied in this paper. Thanks again for pointing out important issues for me.

Comments 5:  In Line 151, the authors have repeatedly mentioned about the input variables that can affect the performance of the model. However, they have not mentioned which are those input variables.

Response 5:  Thank you for your careful review and valuable comments. The mention of input variables that you pointed out does require more explicit elaboration so that the reader can better understand the impact of these variables on model performance. In this paper, relevant review suggestions have been integrated, and variables and their connections have been fully explained in Section 2.1.

Comments 6:  Figure 1 shown in the manuscript is very generalized. If possible, try to make it more specific as per this study so that readers can quickly get the insight. For example, mention the parameters included and briefly describe the significance of each step.

Response 6:  First, I would like to thank you for your review of Figure 1 in the manuscript and for your suggestions. Your professional advice is very valuable to us and we have taken your feedback seriously. After much thought, we decided to keep Figure 1 in its current form without further detailing it. Our consideration is based on the following points: 1. Brevity: We aim to let Figure 1 serve as a quick guide to help the reader get an initial idea of the overall flow of the study. Too much detail may complicate the chart and distract the reader from the core of the study. 2. Focus: We want the reader to quickly grasp the main steps and methodology of the study, rather than getting bogging down in specific parameter details. These details will be discussed in detail in subsequent sections. 3. Overall coordination: Figure 1 is designed to coordinate with the rest of the article and provide a macro view. We believe that the present Figure 1, while remaining concise, has effectively communicated the framework of the study. We believe that the reader can gain a deeper understanding by discussing these parameters and steps in detail in the main text, while Figure 1 serves as an overview to guide the reader to a deeper level of the paper. We are grateful for the opportunity you have given us to set out our considerations and hope that you will understand our explanation.

Comments 7:   In Table 1, what does PLA signify? Does it mean % of PLA material??

Response 7:  Thank you for asking, your question is very accurate. In Table 1, PLA stands for "Polylactic Acid," a biodegradable material widely used in 3D printing. To ensure clarity, we add the meaning of PLA in lines 161-168, which stands for "percentage content of polylactic acid material". We thank you for your careful review, which helped to improve the accuracy and readability of our paper.

Comments 8:   Kindly include the units for all the parameters in Table 1.

Response 8:  Thank you for your advice. I agree with you very much that it is really lack of science not to add units. We have added units in table1. In addition, Twin-screw extrusion experimental conditions is a categorical variable without units.

Comments 9:   The methodology employed in generating the thermal map of the correlation coefficient has to be explained in detail. How the correlation between the PLA and warpage are determined? Is it by analyzing the experimental data?

Response 9:  Thank you for your advice. I agree with you very much. This part is indeed very vague and lacks preciseness. I have made a large adjustment in the article 187-204 lines, and the content of correlation is explained in detail, marked in red. We greatly appreciate your careful review and helpful suggestions, which will help to improve the quality of our paper.

Comments 10:  In Line 186, the hexagonal contour plot, as shown in Figure 2 should be corrected to Figure 3.

Response 10:  Thank you for your reminder. We have revised and marked in red in line 211 of the article.

Comments 11:  The authors should use specific terminologies for readers to better understand without confusion. I can see that the impact strength is also termed as breaking strength, Spline warpage as volume deformation rate. Please use a single terminology for each parameter. 

Response 11:  Thank you very much for your suggestion that the description of the concept has been standardized in the full text to avoid confusion.

Reviewer 3 Report

Comments and Suggestions for Authors

1.      Please correct the missing space before the citations, such as used[1] (line# 43), etc[2] (line# 48), and in the entire manuscript.

2.      A full definition is not provided in the first use of SVR, GBDT, XGB (line#23) and AM (line# 70).

3.      The value of high correlation coefficients for the parameters PLA content, elasticity modulus, and breaking strength is not provided in line#190. For example, PLA content (0.82).

4.      Please briefly explain how authors defined high correlation coefficients, such as higher than 0.7 or 0.8 and provide the citations.

5.      What causes the weak correlations for other parameters? Please briefly explain.

6.      Error in non-superscript of Breaking strength unit kJ/m2 (line#202)

7.      What is the meaning of bracket after the heatmap()? Typo or missing some word/number? Please check.

8.      Please check the underscore in lines #267,268, 2689. Is a typo or typical command used in the model?

9.      Table 4 is not cited in the text.

10.  The 'Python' is not standardized. Some are capital letters (line#436), and some are small letters (line# 378).

11.  Validation using the experimental data is essential to prove the model's reliability. How do authors validate and verify the prediction with the actual experimental results after obtaining the predicted model? 

Author Response

Comments 1:  Please correct the missing space before the citations, such as used[1] (line# 43), etc[2] (line# 48), and in the entire manuscript.

Response 1:  Thank you for your suggestions. I have made modifications in relevant positions in the full text.

Comments 2:  A full definition is not provided in the first use of SVR, GBDT, XGB (line#23) and AM (line# 70).

Response 2: Thank you for pointing this out. I agree with this comment. Therefore, I have adjusted this, marked in red on line 22-34.

Comments 3:The value of high correlation coefficients for the parameters PLA content, elasticity modulus, and breaking strength is not provided in line#190. For example, PLA content (0.82).

Response 3: Thank you for pointing this out.  I think it makes sense. More specific and accurate, it has been added in line 209-210 of the article, marked in red.

Comments 4:Please briefly explain how authors defined high correlation coefficients, such as higher than 0.7 or 0.8 and provide the citations.

Response 4: Thank you for pointing this out. I agree with this comment. Relevance definitions and citations are added in lines 183-188 of the paper.

Comments 5:What causes the weak correlations for other parameters? Please briefly explain.

Response 5: Thank you for raising the question about parameter correlation. In our study, the weak correlation between ADR 4468 chain extender, double helix blending experiment, extrusion swelling ratio and warping may be caused by the following factors: 1. Physical significance of parameters: These parameters are mainly related to the processing properties of the material, rather than directly related to the final mechanical properties of the product, such as warping. Therefore, their effect on warping may be less direct and significant than that of other parameters, such as PLA content and elastic modulus. 2. Data range and distribution: We examined the data for these parameters and found a relatively small range of variation in them, which may have limited their relevance to warping. Dominant role of other variables: PLA content and elastic modulus are strongly correlated with warping in our data, which may mask the correlation of other parameters. 3. Statistical methods: We confirmed that the statistical methods used were suitable for our data analysis and that consistent results were obtained. We believe that the weaker correlation of these parameters may be due to the fact that their role in the 3D printing process is not as directly related to warpage as the other parameters. We will continue to explore other potential effects of these parameters on print results and further explore them in future research. Thank you for your valuable comments and we look forward to your further guidance.

Comments 6:Error in non-superscript of Breaking strength unit kJ/m2 (line#202)

Response 6: Thank you for pointing this out. I have changed it on Line 216.

Comments 7:What is the meaning of bracket after the heatmap()? Typo or missing some word/number? Please check.

Response 7: Thank you for pointing this out. heatmap() is a python function, which is not clear enough. I have changed this sentence to: The correlation coefficient values between input and output parameters were generated using the Seaborn heatmap function, which creates a heatmap by describing the correlation matrix between inputs and outputs.

Comments 8:Please check the underscore in lines #267,268, 2689. Is a typo or typical command used in the model?

Response 8: Thank you for pointing this out. I have removed these.

Comments 9:  Table 4 is not cited in the text.

Response 9: Thank you for pointing this out. I agree with this comment. Therefore, I refer Figure 4 to line 434 in the text and it is marked in red.

Comments 10:  The 'Python' is not standardized. Some are capital letters (line#436), and some are small letters (line# 378).

Response 10: Thank you for pointing this out. I agree with this comment. Therefore, I have changed Python to python on line 205,457,466.、

Comments 11: Validation using the experimental data is essential to prove the model's reliability. How do authors validate and verify the prediction with the actual experimental results after obtaining the predicted model?

Response 11: First of all, we sincerely thank you for your attention to our research work and your valuable comments. Your question about model validation is very pertinent, and experimental validation is indeed a key step in assessing model reliability. In our study, we attach great importance to the experimental verification work of the model. After obtaining the prediction model, we took the following steps to validate and validate the predictions using actual experimental results: 1. Data collection: We first ensured that we collected a series of experimental data relevant to the predictions of the model. The data were obtained under controlled experimental conditions to ensure that it matched the conditions predicted by the model. 2. Model prediction: Using the established model to predict the results of the experiment, we used the same parameters and conditions as the experiment to ensure the accuracy of the prediction. 3. Comparison of results: We compare and analyze the prediction results of the model with the actual collected experimental data. This step involves statistical methods, such as coefficient of determination (R²), root mean square error (RMSE), etc., to quantify the accuracy of model predictions. 4. Sensitivity analysis: We conducted a sensitivity analysis to assess the impact of the uncertainty of the model input parameters on the forecast results, thus further verifying the robustness of the model. 5. Validation report: We documented the validation process in detail and reported the validation results in the paper, including charts and statistical analyses comparing the predicted results with the experimental data.

We recognize that despite the steps we have taken above, there may still be room for improvement. Therefore, we plan to further increase the number and diversity of experimental data in subsequent studies to improve the comprehensiveness and reliability of model validation. We believe that through these detailed validation steps, we can fully demonstrate the reliability of the model and provide a solid foundation for subsequent research and application. Thank you for your suggestions and we look forward to your further guidance.
